# GSK-3β in Pancreatic Cancer: Spotlight on 9-ING-41, Its Therapeutic Potential and Immune Modulatory Properties

**DOI:** 10.3390/biology10070610

**Published:** 2021-07-01

**Authors:** Robin Park, Andrew L. Coveler, Ludimila Cavalcante, Anwaar Saeed

**Affiliations:** 1Department of Medicine, MetroWest Medical Center, Tufts University School of Medicine, Framingham, MA 01702, USA; robin.park@mwmc.com; 2Department of Medicine, Division of Oncology, University of Washington, Seattle, WA 98109-1024, USA; acoveler@uw.edu; 3Actuate Therapeutics, Inc., Fort Worth, TX 76107, USA; ludi.cavalcante@gmail.com; 4Department of Medicine, Division of Medical Oncology, Kansas University Cancer Center & Research Institute, Kansas, KS 66205, USA

**Keywords:** glycogen synthase kinase-3 beta, pancreatic ductal adenocarcinoma, immunotherapy, 9-ING-41

## Abstract

**Simple Summary:**

Glycogen synthase kinase-3 beta is a protein kinase implicated in the promotion and development of various cancers, including pancreatic cancer. In cell culture and animal studies, drugs targeting the inhibition of this protein show treatment potential in pancreatic cancer. Studies show targeting this protein for treatment may overcome resistance to conventional chemotherapy in pancreatic tumors. Early-stage clinical trials are currently studying small molecule inhibitors targeting glycogen synthase kinase-3 beta and interim results show favorable results. Recent studies also suggest that targeting this protein will create synergy with immunotherapy, such as checkpoint inhibitors. Future studies should aim to study new combination treatments involving glycogen synthase kinase-3 beta targeting drugs with chemotherapy and immunotherapy in pancreatic cancer.

**Abstract:**

Glycogen synthase kinase-3 beta is a ubiquitously and constitutively expressed molecule with pleiotropic function. It acts as a protooncogene in the development of several solid tumors including pancreatic cancer through its involvement in various cellular processes including cell proliferation, survival, invasion and metastasis, as well as autophagy. Furthermore, the level of aberrant glycogen synthase kinase-3 beta expression in the nucleus is inversely correlated with tumor differentiation and survival in both in vitro and in vivo models of pancreatic cancer. Small molecule inhibitors of glycogen synthase kinase-3 beta have demonstrated therapeutic potential in pre-clinical models and are currently being evaluated in early phase clinical trials involving pancreatic cancer patients with interim results showing favorable results. Moreover, recent studies support a rationale for the combination of glycogen synthase kinase-3 beta inhibitors with chemotherapy and immunotherapy, warranting the evaluation of novel combination regimens in the future.

## 1. Introduction

Pancreatic ductal adenocarcinoma (PDAC) arises from the epithelial cells of the pancreatic exocrine glands and is the most common malignant tumor of the pancreas. In the United States, nearly 60,000 new cases per-year and 48,000 deaths per-year are attributed to PDAC which comprise about 3% of all new cancer cases and 7% of all cancer-related deaths [1]. Unfortunately, improvement in the prognosis of PDAC has been stagnant compared to other cancers and currently, the 5-year survival rate is 3% for metastatic disease and 10% for all stages combined [2]. Among the newly diagnosed cases, only 20% are eligible for curative resection whereas the rest of the patients are limited to palliative systemic chemotherapy [3].

Currently, the standard of care first-line therapy for metastatic PDAC is limited to gemcitabine plus nab-paclitaxel or FOLFIRINOX (folinic acid, fluorouracil, irinotecan, oxaliplatin) [4]. Gemcitabine with nab-paclitaxel demonstrated improved median overall survival (OS) (8.5 vs. 6.7 mos, HR 0.72, 0.62–0.83) as well as disease control rate (DCR) (48 vs. 33%, RR 1.46, 1.23–1.72) over gemcitabine alone. Furthermore, FOLFIRINOX demonstrated improved median overall survival compared to gemcitabine alone (11.1 vs. 6.8 mos, HR 0.57, 0.45–0.73) in the landmark phase III trial [5]. Additionally, certain targeted therapies have demonstrated incremental survival benefits and retrospective database analyses suggest potential benefit of immunotherapy in PDAC [6,7]. Nonetheless, in absolute terms, the clinical benefit of these regimens is largely disappointing and thus novel strategies are needed to change the course of disease.

Glycogen synthase kinase-3 is a serine/threonine kinase initially discovered as a key regulator of cellular metabolism especially in glycogen biosynthesis [8]. Two different isoforms of glycogen synthase kinase-3 have been described—glycogen synthase kinase-3 alpha (GSK-3α) and glycogen synthase kinase-3 beta (GSK-3β) [8,9]. While GSK-3α and GSK-3β are both highly conserved, ubiquitously expressed, and have many shared substrates, they also have non-overlapping target proteins as well as distinct, non-redundant cellular function. Furthermore, glycogen synthase kinase-3 has over 100 known substrates, which underscores the wide range of effect on various intracellular processes. Likely owing to this reason, genetic deficiency of GSK-3β cannot be complemented by GSK-3α [9]. Since their discoveries, GSK-3α and GSK-3β have been implicated in various disease processes including neurodegenerative disease, chronic inflammatory disorders, and neoplastic processes [10]. However, contemporary research has focused mostly on GSK-3β in the context of cancer, especially in PDAC.

Initially, GSK-3β was thought to act as a tumor suppressor. The reason for this was because of its ability to target several well-known protooncogenic proteins such as beta catenin, cyclin D1, c-Jun, and c-Myc for ubiquitin-mediated proteasomal degradation. However, a growing body of research has established the role of GSK-3β as a protooncogenic protein itself, particularly in the context of ovarian cancer, acute leukemias, glioblastoma, neuroblastoma, and PDAC [11,12,13,14]. Given the pleiotropic nature of GSK-3β, the net effect of GSK-3β perturbance on tumor development is likely context dependent. Nonetheless, GSK-3β has been found to contribute to tumorigenesis in the context of PDAC by promoting cell proliferation, survival, and invasion, metastasis, and chemotherapy resistance. Moreover, recent studies have elucidated the immunomodulatory properties of GSK-3β in the context of cancer, most importantly in regulating negative immune checkpoint molecules but also in directly enhancing the activity of innate and adaptive immune cells as it will be described in later sections.

Here, we review the preclinical studies to date that have established the role of GSK-3β in PDAC tumorigenesis, the early phase clinical trials that have studied pharmacological GSK-3β inhibitors in PDAC and establish a biological rationale for incorporating GSK-3β inhibitors in combination regimens with immunotherapy and chemotherapy to improve PDAC treatment.

## 2. Mechanisms of Altered Regulation of GSK-3β in PDAC

Likely because of its wide-ranging effects, there are multiple layers of regulatory mechanisms to control GSK-3β activity. The physiologic regulation of GSK-3β occurs at both pre-translational and post-translational levels. K-ras mutations are the most common and early driver mutations found in pancreatic cancer. Overexpression of constitutively active Ras isoforms can enhance GSK-3β promoter activity and lead to increased transcription via MAPK signaling [15]. Additionally, the role of micro-RNAs in the transcriptional regulation of GSK-3β in PDAC has also been suggested. Namely, the negative regulation of GSK-3β and the Wnt/beta catenin pathway by microRNA-940 may be involved in the development and progression of PDAC [16,17]. Furthermore, GSK-3β has been shown to modulate epigenetic modification in various settings including DNA methylation during genetic imprinting in embryonic stem cells, and histone modification in the regulation of cell proliferatioin [18,19].

Post-translational modification of GSK-3β has been extensively studied and is relevant in the context of PDAC tumorigenesis. First, protein phosphorylation is a major mechanism of GSK-3β regulation. Phosphorylation at the N-terminal S9 by upstream kinases leads to inactivation of GSK-3β kinase activity, whereas phosphorylation of the Y216 residue in the kinase domain is associated with enhanced kinase activity. Notably, pancreatic cancer cell lines characteristically have increased GSK-3β Y216- and decreased S9-phosphorylation, suggesting that GSK-3β is largely in an activated state in PDAC to drive various pro-tumorigenic processes [20,21,22].

Second, intracellular compartmentalization appears to be also an important mechanism of regulation, especially in the context of cancer. Unlike most protooncogenic kinases studied, GSK-3β is ubiquitously and constitutively expressed in physiologic conditions. In contrast, the expression of GSK-3β in the nucleus is not normally found in healthy normal tissues but is found in abnormal neoplastic cells [10]. The mechanistic importance of this aberrant nuclear localization of GSK-3β is underscored by the fact that target proteins involved in various cell processes linked to tumorigenesis resides in the nucleus, including NF-kB, NFAT, c-Myc, and c-Jun. Although the mechanisms underlying the aberrant nuclear localization of GSK-3β is not fully understood, several studies have elucidated several related pathways. The activation of the PKB/AKT pathway, binding of the protooncogene FRAT1, as well as an N-terminal nuclear localization signal may be linked to this aberrant GSK-3β compartmentalization [23,24,25].

Third, sequestration within multi-protein complexes is another important mechanism of post-translational GSK-3β regulation. Physiologically, GSK-3β is retained in the cytosol as part of a multi-protein complex which includes among other proteins, beta catenin and adenomatous polyposis coli (APC). Binding of Wnt to its cognate receptor Frizzled ultimately results in the dissociation of beta catenin from this multi-protein complex [9,26]. The concurrent regulation of GSK-3β and Wnt/beta catenin signaling and the state of each pathway in PDAC remains poorly understood.

## 3. Role of GSK-3β in PDAC Tumorigenesis

### 3.1. Cell Growth and Proliferation

Several in vitro and in vivo studies have shown that the inhibition of GSK-3 activity can inhibit cell cycle progression and cell proliferation. Inhibition of GSK-3β activity using a small molecule inhibitor led to a reduction in cyclin D1 and cyclin D1/cyclin-dependent kinase (CDK) 4/6 complex-mediated Rb phosphorylation [27]. Furthermore, lithium, which inhibits GSK-3β kinase activity, halts cell cycle progression at the G1/S checkpoint by targeting GLI1 for proteasomal degradation and results in suppressed pancreatic cancer cell proliferation [28]. Treatment of pancreatic tumor xenograft mice with AR-A014418, a selective small molecule GSK-3β inhibitor, causes suppression of tumor growth [29,30]. Additionally, GSK-3β has been found to lead to phosphorylation of NFATc2, resulting in stabilization of nuclear NFATc2 protein levels and its increased transcriptional activity, which in turn promotes cell proliferation, survival, inflammation, and invasion [13,31]. Studies involving pancreatic cancer PDX mice showed that LY2090314, a specific small-molecule GSK-3β inhibitor, led to a reduction in TAK1 and YAP/TAZ protein levels and a consequent decrease in cell proliferation [32]. Indeed, GSK-3β appears to upregulate cell proliferation and prolong survival of PDAC cells.

### 3.2. Cell Invasion and Metastasis

GSK-3β is implicated in cell invasion and metastasis in pancreatic carcinogenesis as its overexpression leads to CXCR4 upregulation and increased MMP-2 expression and results in increased invasiveness of pancreatic cancer cells [33]. GSK-3β inhibition in pancreatic cancer cells leads to perturbations in Rac1 and F-actin subcellular localization, reduced production of MMP-2, and decreased phosphorylation of FAK, leading to inhibition of cell migration and invasion. [27]. Additionally, overexpression of miR-744 and miR-940, which are microRNAs capable of targeting and downregulating GSK-3β mRNA, was found to inhibit cancer cell invasion [16,34]. Thus, increased activity of GSK-3β is associated with increased PDAC invasiveness and metastatic potential while decreased activity is associated with decreased and inhibited invasion.

Epithelial-to-mesenchymal transition (EMT) is the acquirement of epithelial cells of a mesenchymal cell phenotype. It is physiologically necessary for various biological processes such as embryonic development and wound healing. However, in the setting of tumorigenesis, cancer cells exploit this cellular mechanism to enhance their own invasiveness and their metastatic potential. Furthermore, EMT is an established hallmark of PDAC. Studies have implicated the role of GSK-3β inhibition in regulating EMT via effects on the Snail/E-cadherin pathway [35]. Further studies are warranted to determine the role of GSK-3β inhibition in PDAC treatment with respect to EMT suppression.

### 3.3. Cell Metabolism

Consistent with its initial discovery as a regulator of cell metabolism, GSK-3β and its impact on cell metabolism is relevant in tumorigenesis. The regulation of autophagy and glycolysis appears to be important in PDAC. Autophagy is a highly regulated cellular process involved in the turnover and recycling of intracellular components. Altered autophagic processes in tumor cells has been associated with tumor progression and altered tumor immunity. Autophagy has recently been gaining attention as a key mechanism and target for anti-tumor therapy [36]. Increased phosphorylation of ULK1 at S405 and S415, which lead to its activation, is found in several human pancreatic cancer cell lines. Moreover, such pancreatic cell lines notably have increased autophagy. Of note, GSK-3β was recently found to phosphorylate ULK1 at S405 and S415, suggesting its role in mediating autophagy in PDAC. Inhibition of ULK1 phosphorylation at these two residues led to decreased pancreatic tumor cell survival, underpinning the reliance of the tumor cells on autophagy [37]. Thus, autophagy may be yet another cellular process that is altered via aberrant GSK-3β activity in PDAC.

The shifting of glucose metabolism from oxidative phosphorylation to glycolysis is a well-known hallmark of cancer cells including PDAC [38]. A transcriptomic profiling study has demonstrated that the loss of an endodermal lineage specification gene (*HNF4A)* upregulates GSK3β and drives a squamous cell-like metabolic profile. Pharmacological GSK-3β targeting inhibits glycolysis in patient derived cell lines from squamous subtype PDAC and results in selective responses [39]. This suggests that targeting of deranged glucose metabolism is another facet of the mechanism of action of GSK3β inhibitors.

## 4. Role of GSK-3β in Chemotherapy Resistance in PDAC

PDAC is well noted for its resistance to conventional cytotoxic anti-cancer therapy. Pancreatic tumors are characterized by fibrosis, desmoplasia, hypovascularity, and concurrent activation of various prooncogenic signaling pathways including pro-survival pathways which contribute to its broad drug resistance [40,41]. GSK-3β is central in mediating this drug resistance mainly via upregulation of pro-survival signaling pathways which ultimately lead to upregulation of pro-survival molecules such as Bcl-2 and XIAP [42].

Pre-clinical models have demonstrated synergy in combining cytotoxic chemotherapy and GSK-3β inhibition. The toolkit GSK-3β inhibitor AR-A014418 demonstrated synergistic cytotoxicity in combination with gemcitabine in PANC-1 cells [43,44]. Furthermore, lithium, which is known for its GSK-3β inhibitory activity, has been shown to enhance the anti-cancer activity of gemcitabine via downregulation of the hedgehog-GLI1 pathway [28]. Additionally, a toolkit inhibitor with dual-inhibitory activity of GSK-3β and histone deacetylase 2 was found to enhance the cytotoxicity of gemcitabine and paclitaxel of PDAC cells [45].

GSK-3β inhibition was found to enhance gemcitabine activity in PDAC PDX mice by preventing the triggering of the ATR DNA damage response pathway. GSK kinase activity is required to prevent the degradation of a key adaptor molecule which is necessary for the full activation of ATR. Ding and colleagues used 9-ING-41 to demonstrate these results which is significant as 9-ING-41 is currently being studied in clinical trials and will be detailed in a later section. In keeping with its effect on the ATR DNA damage response pathway, 9-ING-41 demonstrated a strong cell cycle arrest inducing effect in vitro in other studies [46,47,48,49].

In addition, the interaction between GSK-3β and therapeutic agents other than chemotherapy have been noted as well as the effects of natural products on GSK-3β [50]. Abrams and colleagues demonstrated that transfection of functional GSK-3β in K-Ras-dependent PDAC cells reduced their sensitivity to chemotherapy as well as various signal transduction inhibitors including those that inhibit EGFR/HER2, ALK/AXL/FLT3, KRAS, MEK1, and mTORC1, suggesting the potential significance of GSK-3β inhibition in conjunction with other small molecule inhibitors [51,52]. Furthermore, several natural products including resveratrol, berberine, and curcumin have been known to modulate GSK-3β activity and have varying effects on in vitro models of colon cancer, melanoma, and breast cancer. Thus, natural products targeting GSK-3β may be a therapeutic avenue worth exploring [53].

GSK-3β inhibition via 9-ING-41 has shown anti-fibrotic effects in in vitro and in vivo models of idiopathic pulmonary fibrosis (IPF). 9-ING-41 was found to significantly inhibit the induction of myofibroblasts in vitro.9-ING-41 attenuated disease progression in TGF-β-induced IPF animal models [54]. These results indicate that GSK-3β inhibition can attenuate the progression of fibrosis. Importantly, these results can be extrapolated to the treatment of PDAC with GSK-3β inhibition given that fibrosis is a hallmark of PDAC which contributes towards chemotherapy resistance.

Thus, there is strong rationale for the combination of GSK-3β inhibition and cytotoxic chemotherapy agents with the aim of overcoming treatment resistance by modulating the DNA damage response elicited by the chemotherapeutic agents. Given the pleiotropic nature of GSK-3β, the directional effect of GSK-3β inhibition on concomitantly administered chemotherapeutic or targeted agent may be unpredictable. Therefore, the importance of rationale, biological mechanism-based treatment combinations will be crucial in identifying beneficial GSK-3β-based combination therapies.

## 5. Pharmacological GSK-3β Inhibition

In pre-clinical studies of PDAC, several pharmacological modalities have been used to target GSK-3β for inhibition including small interfering RNAs as well as small molecule inhibitors. Most of the focus in translating GSK-3β inhibitors for PDAC to the clinic has been on small molecule inhibitors. While many small molecule GSK-3β inhibitors have been developed in this setting, most were not further evaluated in clinical trials because of unfavorable pharmacokinetic properties, making them unfavorable for human use.

Two small molecule GSK-3β inhibitors have made their way into clinical trials. LY2090314 is a potent, selective small molecule inhibitor of GSK-3β. Unfortunately, the phase I trial in advanced PDAC patients for this agent was opened but was terminated prematurely due to a short half-life and poor pharmacokinetic performance of the drug (Table 1).

9-ING-41 is also a potent, selective small molecule inhibitor of GSK-3β, which has demonstrated anti-tumor activity in patient-derived xenograft mice (PDX) of various human cancers including PDAC [49,55,56,57,58,59]. Its inhibitory effects on NF-kB with its consequent suppression of downstream NF-kB target genes such as cyclin D1, Bcl-2, anti-apoptotic protein (XIAP) and B-cell lymphoma-extra large (Bcl-XL) appear to be important in mediating the anti-tumor activity. 9-ING-41 enhances the effects of various chemotherapeutic agents on cell growth suppression in pancreatic cancer cell cultures [42]. Furthermore, it enhances the anti-tumor activity of gemcitabine in PDX models of pancreatic cancer. The TopBP1/ATR/Chk1 DNA damage response pathway appears to be implicated in this chemo-potentiating activity. Treatment of pancreatic PDX mice with 9-ING-41 prevents gemcitabine-induced cell cycle arrest by impairing the activation of ATR, leading to the consequent activation of Chk1 [46].

## 6. Clinical Experience with Small Molecule GSK-3β Inhibitors in PDAC

9-ING-41 is currently being studied in a phase Ib/II basket trial of treatment-refractory advanced solid tumors or hematologic malignancies (NCT03678883, Actuate 1801). The interim results of the study were presented at the 2020 American Society of Clinical Oncology Virtual Annual Conference. The study is composed of parts 1 and 2 which assess safety and dose-limiting toxicities, as well as the recommended phase II dose for 9-ING-41 monotherapy (part 1) or in combination with chemotherapy (part 2). Part 3 is studying clinical activity of 9-ING-41 in combination with chemotherapy. At the time of data cutoff, a total of 155 patients were evaluable including 42 patients with PDAC who had received more than 1 dose of study drug (Figure 1) [60].

In part 1 of the 1801 study, 67 patients received 9-ING-41 on a twice weekly IV schedule at eight dose levels: 1.0, 2.0, 3.3, 5, 7, 9.3, 12.4 and 15 mg/kg. No 9-ING-41-attributable SAEs were observed. The dose of 15 mg/kg has been deemed the RP2D based on the volume of fluid given with this dose.

Furthermore, grades 1 and 2 transient visual changes characterized by changes in color perception during infusion have been observed starting at 3.3 mg/kg and are indicative of target engagement as GSK-3Β can be found within photoreceptors. These symptoms lasted up to several hours but were completely reversible. No attendant abnormalities have been demonstrable on retinal or general ocular examinations either.

There were also grades 1 and 2 infusion reactions associated with 9-ING-41 in a minority of patients. These resolved upon rapid institution of local infusion reaction protocols, usually involving administration of anti-pyretics, anti-histamines, intravenous fluids and occasionally steroids. Patients who presented with infusion reactions were required to pre-medicate with the same medications upon subsequent infusions.

Currently, 1 patient remains on Part 1 of the study with refractory *BRAF^V600K^* mutated melanoma with central nervous system disease in complete response after 2 years, and another patient with adult T-cell leukemia/lymphoma had a sustained partial response of over 15 months.

In part 2 of the study, 171 patients with advanced solid tumors have received 9-ING-41 on a twice weekly schedule at doses of 3.3, 5, 7, 9.3, 12.4 and 15 mg/kg combined with carboplatin, doxorubicin, gemcitabine, irinotecan, lomustine, carboplatin plus paclitaxel, carboplatin plus pemetrexed or gemcitabine plus nab-paclitaxel. Patients initially needed to have been previously exposed to that same chemotherapy agent to qualify for study entry, but this was later amended to allow first-line doxorubicin patients. One 9-ING-41 attributable grade 3 SAE of visual disturbance was observed, but no other clinically significant AEs attributable to the investigational product have been noted in this cohort. The RP2D for all combinations was determined to be 15 mg/kg IV. Six patients remain on part 2 of the study, and clinical responses include sustained partial responses in PDAC, endometrial cancer, cervical cancer and undifferentiated pleomorphic sarcoma, as well as prolonged disease stability in several histologies, including PDAC, appendiceal cancer and mesothelioma.

Part 3 of the study will be a phase II signal-seeking expansion cohort where advanced PDAC patients will receive gemcitabine/nab-paclitaxel plus 9-ING-41 for first-line treatment.

## 7. Immunomodulation via GSK-3β Inhibition

Immunotherapy has shifted the paradigm of anti-cancer treatment in solid tumors. Namely, immune checkpoint inhibitors targeting the programmed death-1 and programmed death ligand-1 (PD-1/PD-L1) pathway have become part of the standard of care treatment in various cancers including melanoma, non-small cell lung cancer [61,62,63,64,65]. Immune checkpoint inhibitors have demonstrated largely disappointing results in PDAC patients thus far. For example, the Canadian PA.7 trial compared durvalumab (anti-PD-L1) and tremelimumab (anti-CTLA-4) with chemotherapy versus chemotherapy alone in first-line metastatic PDAC in a phase II setting. The study showed no significant difference in the primary endpoint of OS as well as the secondary endpoints of PFS and ORR [66]. The immune suppressive tumor microenvironment along with extensive desmoplasia of pancreatic tumors have resulted in this poor responsiveness to immunotherapy [67,68]. Novel combination therapeutic approaches aimed at re-invigorating the tumor microenvironment such that it is favorable for the elicitation of anti-tumor immune responses are needed to overcome such treatment barriers to immunotherapy.

### 7.1. Regulation of Immune Checkpoint Molecule Expression

The immunomodulatory role of GSK-3β has been demonstrated in vitro and in vivo. GSK-3β appears to be associated with the expression of immune checkpoint molecules (Figure 2). GSK-3β was found to be an upstream regulator of the transcriptional activation of PD-1 on T cells [69]. Mechanistically, inhibition of GSK-3β appears to upregulate the expression of the T-bet, which is a transcription factor required for the differentiation of T cells into T helper type 1 cells [70]. Furthermore, suppression of GSK-3β activity using small molecule inhibitors led to reduced levels of PD-1 and enhanced activity of cytotoxic T cells [71]. The reduction in PD-1 levels, enhancement of cytotoxic T cell activity via GSK-3β inhibition has been seen in the context of various syngeneic mouse models including pancreatic cancer models [72]. Inhibition of GSK-3β has the potential to reinvigorate exhausted cytotoxic T cells in the immune suppressive tumor microenvironment of pancreatic tumors.

The effect of GSK-3β on immune checkpoint regulation is not limited to PD-1. Lymphocyte activation gene-3 (LAG-3) is an activation antigen that also negatively regulates T cells. Notably, LAG-3 and PD-1 are co-expressed at high levels in tumor infiltrating lymphocytes in certain tumor model mice and the inhibition of both molecules enhances CD8+ effector T cell numbers [73,74,75]. Furthermore, LAG-3 expression has been noted in tumor-infiltrating T cells in tumor specimens from PDAC patients. The expression of LAG-3 inversely correlates with disease free survival, suggesting the importance of this marker in PDAC patients [76]. GSK-3β negatively regulates the cell surface expression of LAG-3 on tumor infiltrating T lymphocytes. The regulation of LAG-3 expression by GSK-3β is mediated via T-bet in the same manner as PD-1 expression. Inhibition of GSK-3β leads to the loss of LAG-3 on T cells and enhanced tumor clearance [77]. The anti-LAG-3 antibody (relatlimab) is currently being evaluated in first-line advanced melanoma and has demonstrated progression free survival benefit when added to anti-PD-1 antibody (nivolumab) [78]. Thus, co-inhibition of LAG-3 and GSK-3β may be a therapeutic avenue worth exploring in PDAC treatment.

The immunomodulatory effects of 9-ING-41 have been demonstrated in the preclinical setting in various tumor cells and animal models. In melanoma mouse models, 9-ING-41 downregulates the expression of inhibitory immune checkpoints including PD-1 and LAG-3, resulting in synergistic anti-tumor activity when administered in combination with immune checkpoint inhibitors [79]. 9-ING-41 also enhanced the cytotoxicity of cytokine activated immune cells on renal cancer cell lines [80] In colorectal cancer cells, 9-ING-41 enhances the anti-tumor activity of NK and T cells [81]. Thus, small molecule inhibition of GSK-3β has immunomodulatory effects which support its therapeutic partnership with immune checkpoint inhibitors.

Interestingly, a complex interaction exists among GSK-3β inhibition and the DNA damage response pathway with immune checkpoint modulation. In xenograft models of breast cancer, poly ADP ribose polymerase (PARP) inhibition was shown to upregulate tumor PD-L1 expression via GSK-3β inactivation, suggesting a potential link between not only GSK-3β with immunomodulation but also with DNA damage response modulation [82]. Similar findings were also seen in colorectal cancer cell lines and model mice, where GSK inhibition enhanced the treatment effect of PARP inhibition [83]. As PARP inhibitors are a topic of great interest in PDAC, further studies are warranted to elucidate the complex interactions at the crossroads of GSK-3β, DNA damage response, and anti-tumor immunity.

### 7.2. Regulation of Immune Cell Function

In addition to the regulation of immune checkpoint expression, GSK-3β is also known to directly regulate the activity of antigen-specific T cells. Studies using transgenic mice have demonstrated augmentation of GSK-3β expression in T cells lead to reduced proliferation and interleukin-2 production. Conversely, inhibition of GSK-3β using lithium enhances T cell proliferation and interleukin-2 production [84]. The inhibition of GSK-3β has been shown to complement the function of CD28 as a costimulatory molecule in the proliferation of human T cells [85] and has been found to prolong the survival and enhance the anti-tumor cytotoxicity of chimeric antigen receptor T cells in the setting of a glioblastoma model [86,87]. GSK-3β inhibition enhances the cytotoxicity of CD8+ T cells in vitro against gastric cancer cells [88]. Taken together, the immunomodulatory effects of GSK-3β inhibition go beyond the regulation of immune checkpoints to the direct enhancement of anti-tumor T cell mediated cytotoxicity.

Similarly, GSK-3β inhibition appears to enhance the maturation, expansion, and cytotoxicity of natural killer (NK) cells. In an ex vivo study, the acquisition of maturation cell markers and proliferation of NK cells were enhanced via the addition of a competitive pan-GSK inhibitor (CHIR99021) to the culture media. The use of CHIR99021-stimulated NK cells as treatments are currently being evaluated in a phase I trial (NCT03319459). CHIR99021 led to greater production of pro-inflammatory cytokines (TNFα, IFNγ) and increased cytotoxicity against a wide range of cancer cell lines including pancreatic cancer cells [89]. Thus, GSK-3β has the potential to directly invigorate tumor infiltrating T cells and NK cells by enhancing proliferative capacity and cytotoxicity.

## 8. Conclusions

In summary, a growing body of evidence supports GSK-3β as a valid therapeutic target in PDAC. The impact of GSK-3β in promoting various prooncogenic processes including cell proliferation, survival, and invasion, metastasis, and autophagy has been well studied. The role of GSK-3β in promoting drug resistance in PDAC by modulating the ATR/TopBP1-mediated DNA damage response pathway supports its rationale for combining with chemotherapy. Despite early setbacks, the results of early phase clinical trials studying small molecule GSK-3β inhibitors are promising with acceptable safety and promising efficacy. Moreover, GSK-3β is a known regulator of negative immune checkpoint expression and of innate and adaptive immune cell proliferation and cytotoxicity. This supports the rationale of combining GSK-3β inhibitors with immune checkpoint inhibitors in the treatment of advanced PDAC. Clinical trials evaluating such novel combinations in PDAC are warranted to assess safety and clinical efficacy. Future studies will also need to explore biomarkers that will enrich patient populations for positive responses to such combination therapies involving GSK-3β inhibitors.

## Figures and Tables

**Figure 1 biology-10-00610-f001:**
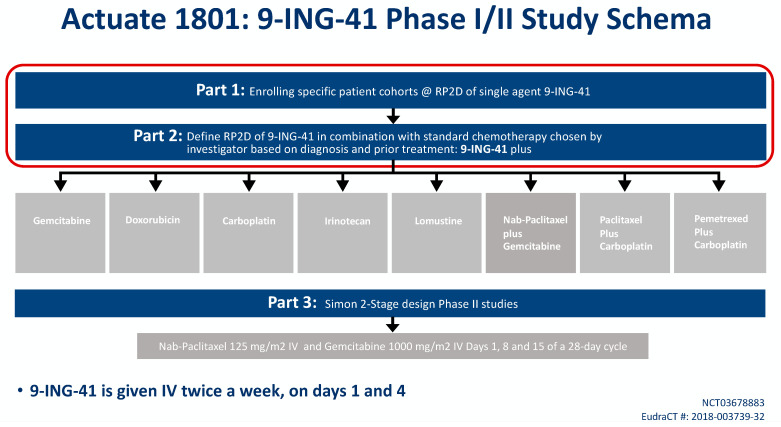
Trial design of the Actuate 1801 trial. The objectives of the ongoing Part 1 and Part 2 are to determine the recommended phase II dose (RP2D) of 9-ING-41 with and without chemotherapy, respectively. The choice of the chemotherapy agent will be chosen by the investigator and depend on the diagnosis and prior treatment. Part 3 will be a Simon 2-stage phase II design which is an expansion cohort that will open based on the results of Part 1 and 2. It will evaluate 9-ING-41 with gemcitabine with nab-paclitaxel in advanced pancreatic ductal cancer (PDAC) for first-line therapy.

**Figure 2 biology-10-00610-f002:**
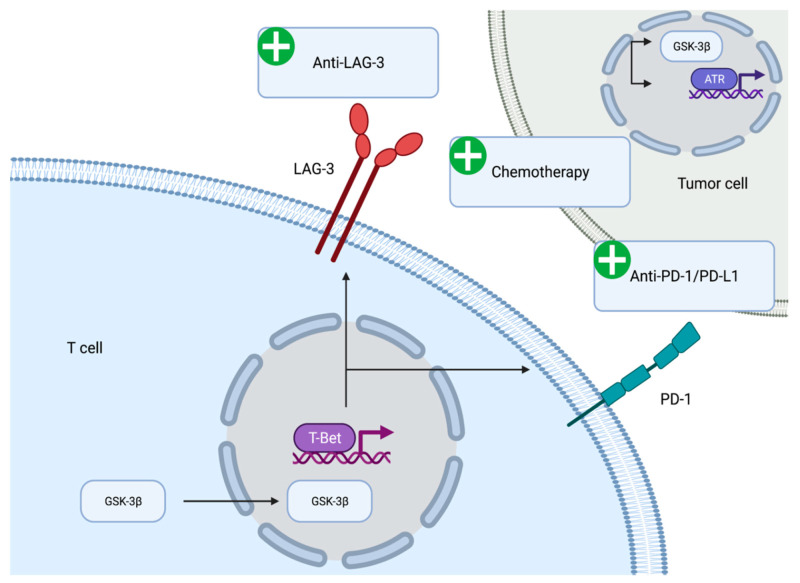
Rationale for combining GSK-3β inhibitors with immunotherapy and chemotherapy. GSK-3β increases tumor expression of programmed death ligand-1 (PD-1) and lymphocyte activation gene-3 (LAG-3), both of which can be reversed with small molecule inhibition of GSK-3β. Furthermore, GSK-3β inhibition can prevent DNA damage response activation by chemotherapeutic agents such as gemcitabine by preventing the induction of the ATR/TopBP1 pathway. Such mechanisms support a rationale for combining GSK-3β small molecule inhibitors with chemotherapy and immunotherapy. Created with BioRender.com (accessed on 21 February 2021).

**Table 1 biology-10-00610-t001:** Ongoing Trials Evaluating GSK-3β Inhibitors in Pancreatic Cancer.

Trial ID/Name	Phase	Treatment Arm	Status
NCT03678883	Ib/II	9-ING-41	Active, recruiting
NCT01632306	I/II	LY2090314	Terminated

## Data Availability

Not applicable.

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
