# Peer review of "GSK-3β in Pancreatic Cancer: Spotlight on 9-ING-41, Its Therapeutic Potential and Immune Modulatory Properties"

_biology, 2021, doi:10.3390/biology10070610_

Round 1

Reviewer 1 Report

Park and colleagues proposed a review article aimed at summarizing the recent findings of the role of GSK-3β in pancreatic cancer and in particular about the therapeutic potential of GSK-3β inhibitors. Overall, the manuscript is a significant improvement of the current knowledge about GSK-3β, however, some parts should be better described. Below are some minor/major comments that will improve the quality of the manuscript:
1) Please be consistent with the use of GSK-3β. Revise it also in the title;
2) In the second paragraph of the Introduction section, please briefly mention the evolution of novel pharmacological treatment with the introduction of targeted therapies and immunotherapies and the slight improvement of the PFS and OS time. For this purpose, please see:
- PMID: 31598392
- PMID: 32613090
3) In Chapter 2, the authors only mentioned the role of microRNAs in the regulation of GSK-3β. Please, better argue this aspect highlighting the miRNAs involved and the molecular pathways altered by miRNA alterations. In addition, please describe any epigenetic phenomena, like DNA methylation or histone modifications regulating GSK-3β expression. For this purpose, see:
- PMID: 25833708
- PMID: 27999207
- PMID: 24704462
4) Chapter 4 should be improved. Please describe how GSK-3β inhibition increase pancreatic cell sensitivity to other chemotherapeutic agents, targeted therapies and nutraceuticals used both in vitro and in vivo for the treatment of PDAC. For this purpose, please see:
- PMID: 32178549
- PMID: 33917370
- PMID: 30975030
- PMID: 22041920
- PMID: 32365809
5) In line 236, the authors indicate “Table 1”, however, no table is showed. Is Table 1 referred to the table shown in line 392?. Please, clarify and, eventually, move in the proper position;
6) Provide references for the following sentences: “Immunotherapy has shifted the paradigm of anti-cancer treatment in solid tumors. Namely, immune checkpoint inhibitors targeting the programmed death-1 and programmed death ligand-1 (PD-1/PD-L1) pathway have become part of the standard of care treatment in various cancers including melanoma, non-small cell lung cancer [52].”. See the above suggestions.

Author Response

1) Please be consistent with the use of GSK-3β. Revise it also in the title;

  • We appreciate the reviewer’s comment. The following comment has been addressed.

2) In the second paragraph of the Introduction section, please briefly mention the evolution of novel pharmacological treatment with the introduction of targeted therapies and immunotherapies and the slight improvement of the PFS and OS time. For this purpose, please see:

- PMID: 31598392

- PMID: 32613090

  • We appreciate the reviewer’s comment. The following has been added to the manuscript: Additionally, certain targeted therapies have demonstrated incremental survival benefits and retrospective database analyses suggest potential benefit of immunotherapy in PDAC [6,7].

3) In Chapter 2, the authors only mentioned the role of microRNAs in the regulation of GSK-3β. Please, better argue this aspect highlighting the miRNAs involved and the molecular pathways altered by miRNA alterations. In addition, please describe any epigenetic phenomena, like DNA methylation or histone modifications regulating GSK-3β expression. For this purpose, see:

- PMID: 25833708

- PMID: 27999207

- PMID: 24704462

  • We appreciate the reviewer’s comment. The following has been added to the manuscript: Namely, the negative regulation of GSK-3β and the Wnt/beta catenin pathway by mi-croRNA-940 may be involved in the development and progression of PDAC [16,17]. Furthermore, GSK-3β has been shown to modulate epigenetic modification in various settings including DNA methylation during genetic imprinting in embryonic stem cells, and histone modification in the regulation of cell proliferatioin [18,19].

4) Chapter 4 should be improved. Please describe how GSK-3β inhibition increase pancreatic cell sensitivity to other chemotherapeutic agents, targeted therapies and nutraceuticals used both in vitro and in vivo for the treatment of PDAC. For this purpose, please see:

- PMID: 32178549

- PMID: 33917370

- PMID: 30975030

- PMID: 22041920

- PMID: 32365809

  • We appreciate the reviewer’s comment. The following has been added to the manuscript: In addition, the interaction between GSK-3β and therapeutic agents other than chemotherapy have been noted as well as the effects of natural products on GSK-3β [50]. Abrams and colleagues demonstrated that transfection of functional GSK-3β in K-Ras-dependent PDAC cells reduced their sensitivity to chemotherapy as well as various signal transduction inhibitors including those that inhibit EGFR/HER2, ALK/AXL/FLT3, KRAS, MEK1, and mTORC1, suggesting the potential significance of GSK-3β inhibition in conjunction with other small molecule inhibitors [51][52]. Furthermore, several natural products including resveratrol, berberine, and curcumin have been known to modulate GSK-3β activity and have varying effects on in vitro models of colon cancer, melanoma, and breast cancer. Thus, natural products targeting GSK-3β may be a therapeutic avenue worth exploring [53].

5) In line 236, the authors indicate “Table 1”, however, no table is showed. Is Table 1 referred to the table shown in line 392?. Please, clarify and, eventually, move in the proper position;

  • We appreciate the reviewer’s comment. Table 1 has been moved to the appropriate location.

6) Provide references for the following sentences: “Immunotherapy has shifted the paradigm of anti-cancer treatment in solid tumors. Namely, immune checkpoint inhibitors targeting the programmed death-1 and programmed death ligand-1 (PD-1/PD-L1) pathway have become part of the standard of care treatment in various cancers including melanoma, non-small cell lung cancer [52].”. See the above suggestions.

  • We appreciate the reviewer’s comment. Additional references cited as recommended:
  1. Larkin, J.; Chiarion-Sileni, V.; Gonzalez, R.; Grob, J.-J.; Rutkowski, P.; Lao, C.D.; Cowey, C.L.; Schadendorf, D.; Wagstaff, J.; Dummer, R.; et al. Five-Year Survival with Combined Nivolumab and Ipilimumab in Advanced Melanoma. New England Journal of Medicine 2019, 381, 1535-1546, doi:10.1056/NEJMoa1910836.
  2. Robert, C.; Ribas, A.; Schachter, J.; Arance, A.; Grob, J.-J.; Mortier, L.; Daud, A.; Carlino, M.S.; McNeil, C.M.; Lotem, M.; et al. Pembrolizumab versus ipilimumab in advanced melanoma (KEYNOTE-006): post-hoc 5-year results from an open-label, multicentre, randomised, controlled, phase 3 study. The Lancet Oncology 2019, 20, 1239-1251, doi:10.1016/S1470-2045(19)30388-2.
  3. Reck, M.; Rodríguez-Abreu, D.; Robinson, A.G.; Hui, R.; Csőszi, T.; Fülöp, A.; Gottfried, M.; Peled, N.; Tafreshi, A.; Cuffe, S.; et al. Pembrolizumab versus Chemotherapy for PD-L1–Positive Non–Small-Cell Lung Cancer. New England Journal of Medicine 2016, 375, 1823-1833, doi:10.1056/NEJMoa1606774.
  4. Borghaei, H.; Paz-Ares, L.; Horn, L.; Spigel, D.R.; Steins, M.; Ready, N.E.; Chow, L.Q.; Vokes, E.E.; Felip, E.; Holgado, E.; et al. Nivolumab versus Docetaxel in Advanced Nonsquamous Non–Small-Cell Lung Cancer. New England Journal of Medicine 2015, 373, 1627-1639, doi:10.1056/NEJMoa1507643.

Reviewer 2 Report

Minor:

Author has presented precise and accurate summery of 9-ING-41 clinic trial and its future use in combination with chemotherapy. However, controlling the activity of GSK3 beta (S9 and Y216) is very tricky because upstream kinases can regulates its in either way. Of note, GSK3 beta signaling cascade is very sensitive to metabolic changes (as author well explained in article itself). How author would like to address the effect of other inhibitors (used in chemotherapy) on GSK3 beta activity?      

Major:

Author has nicely covered the brief history and implication of GSK3 beta inhibitors in different biological fields. Author gave the good perspective of using 9-ING-41 in combination of immunotherapy. However, using GSK3 beta inhibitors in combination with immunotherapy is a long odd and need top tier scrutiny.

Author Response

Minor:

1) Author has presented precise and accurate summery of 9-ING-41 clinic trial and its future use in combination with chemotherapy. However, controlling the activity of GSK3 beta (S9 and Y216) is very tricky because upstream kinases can regulates its in either way. Of note, GSK3 beta signaling cascade is very sensitive to metabolic changes (as author well explained in article itself). How author would like to address the effect of other inhibitors (used in chemotherapy) on GSK3 beta activity?     

Major:

  • We appreciate the reviewer’s comment. The following has been added: Given the pleiotropic nature of GSK-3β, the directional effect of GSK-3β inhibition on concomitantly administered chemotherapeutic or targeted agent may be unpredictable. Therefore, the importance of rationale, biological mechanism-based treatment combinations will be crucial in identifying beneficial GSK-3β-based combination therapies.

2) Author has nicely covered the brief history and implication of GSK3 beta inhibitors in different biological fields. Author gave the good perspective of using 9-ING-41 in combination of immunotherapy. However, using GSK3 beta inhibitors in combination with immunotherapy is a long odd and need top tier scrutiny.

  • We appreciate the reviewer’s comment. The authors agree that combining GSK-3b inhibitors with immunotherapy is experimental and requires further validation in early phase clinical trials. Nonetheless, we believe there is a strong rationale for considering this combination given the recently discovered immunomodulatory effects of GSK-3b inhibition as described in our review.